# Revisiting the Radiosynthesis of [^18^F]FPEB and Preliminary PET Imaging in a Mouse Model of Alzheimer’s Disease

**DOI:** 10.3390/molecules25040982

**Published:** 2020-02-22

**Authors:** Cassis Varlow, Emily Murrell, Jason P. Holland, Alina Kassenbrock, Whitney Shannon, Steven H. Liang, Neil Vasdev, Nickeisha A. Stephenson

**Affiliations:** 1Azrieli Centre for Neuro-Radiochemistry, Brain Health Imaging Centre, Centre for Addiction and Mental Health, Toronto, ON M5T 1R8, Canada; cassis.varlow@mail.utoronto.ca (C.V.); emily.murrell@camhpet.ca (E.M.); wes728@mail.usask.ca (W.S.); 2Institute of Medical Science, University of Toronto, Toronto, ON M5S1A8, Canada; 3Division of Nuclear Medicine and Molecular Imaging, Massachusetts General Hospital & Department of Radiology, Harvard Medical School, Boston, MA 02114, USA; jason.holland@chem.uzh.ch (J.P.H.); alinakassenbrock@gmail.com (A.K.); Liang.Steven@mgh.harvard.edu (S.H.L.); 4Department of Chemistry, University of Zurich, 8057 Zurich, Switzerland; 5Department of Chemistry, University of Saskatchewan, Saskatoon, SK S7N OX2, Canada; 6Department of Psychiatry, University of Toronto, Toronto, ON M5T-1R8, Canada; 7Department of Chemistry, The University of West Indies at Mona, Kingston 7, Jamaica

**Keywords:** [^18^F]FPEB, mGluR5, positron emission tomography (PET), iodonium-ylide, Alzheimer’s Disease (AD)

## Abstract

[^18^F]FPEB is a positron emission tomography (PET) radiopharmaceutical used for imaging the abundance and distribution of mGluR5 in the central nervous system (CNS). Efficient radiolabeling of the aromatic ring of [^18^F]FPEB has been an ongoing challenge. Herein, five metal-free precursors for the radiofluorination of [^18^F]FPEB were compared, namely, a chloro-, nitro-, sulfonium salt, and two spirocyclic iodonium ylide (SCIDY) precursors bearing a cyclopentyl (SPI5) and a new adamantyl (SPIAd) auxiliary. The chloro- and nitro-precursors resulted in a low radiochemical yield (<10% RCY), whereas both SCIDY precursors and the sulfonium salt precursor produced [^18^F]FPEB in the highest RCYs of 25% and 36%, respectively. Preliminary PET/CT imaging studies with [^18^F]FPEB were conducted in a transgenic model of Alzheimer’s Disease (AD) using B6C3-Tg(APPswe,PSEN1dE9)85Dbo/J (APP/PS1) mice, and data were compared with age-matched wild-type (WT) B6C3F1/J control mice. In APP/PS1 mice, whole brain distribution at 5 min post-injection showed a slightly higher uptake (SUV = 4.8 ± 0.4) than in age-matched controls (SUV = 4.0 ± 0.2). Further studies to explore mGluR5 as an early biomarker for AD are underway.

## 1. Introduction

*L*-Glutamate is the primary excitatory neurotransmitter at the majority of excitatory synapses in the mammalian central nervous system (CNS). Signaling processes involving metabotropic glutamate occur via membrane-bound G-protein coupled receptors (GPCRs) known as metabotropic glutamate receptors (mGluRs). There are eight mGluR subtypes, of which the mGluR5 subtype is involved in the excitatory signaling cascade of intracellular calcium release, playing a vital role in brain development, learning, memory, and in maintaining synaptic plasticity [1]. Changes in mGluR5 expression have been identified in several neuropathological diseases and disorders including addiction, Parkinson’s disease, post-traumatic stress disorder (PTSD), epilepsy, Huntington’s disease, and Alzheimer’s disease (AD), making mGluR5 an attractive target for the development of new therapeutics and for monitoring the progression of several CNS disease states [2,3,4]. Positron emission tomography (PET) serves as a highly sensitive and non-invasive means of monitoring the changes in mGluR5 distribution and regulation associated with pathophysiological conditions in the CNS [5,6]. [^18^F]3-Fluoro-5-[(pyridin-3-yl)ethynyl] benzonitrile ([^18^F]FPEB) was developed as a mGluR5 PET ligand and is widely used in clinical research [7,8].

The routine radiosynthesis of [^18^F]FPEB has traditionally been low yielding because nucleophilic aromatic substitution by [^18^F]fluoride is not favored as the nitrile group is positioned meta to the leaving group on the aromatic ring of the precursor. Furthermore, high temperature reactions are generally required, leading to decomposition products and radiochemical impurities. [^18^F]FPEB is commonly synthesized from a chloro- or nitro-precursor in low radiochemical yields (RCYs, <10%) as shown in Table 1 (1 and 2, respectively). When initially reported by Merck Research Laboratories, [^18^F]FPEB was prepared via a S_N_Ar reaction of an aryl-chloro precursor (**1**) with labeled potassium cryptand fluoride, [K_222_][^18^F], and K_2_CO_3_ as the base [9]. This reaction resulted in 5% RCY and required microwave conditions. Like the majority of others, our development of [^18^F]FPEB for human use was initiated with the commercially available nitro-precursor and a low RCY (4%) was achieved [10]. Compound **3** is a boronic acid precursor and **4** is an arylstannane precursor that were used in copper-mediated radiofluorinations to synthesize [^18^F]FPEB in decay-corrected automated radiochemical conversions (RCC; not isolated) of 5% and 11% respectively [11,12]. Translating these precursors for the radiopharmaceutical production of [^18^F]FPEB will be challenging as lower RCY can be expected upon purification and formulation, while additional testing for residual metals will be required.

In our efforts to optimize the efficiency of ^18^F-C_sp2_ bond formation, we discovered that the use of harsh reaction conditions such as high temperatures and base concentrations resulted in the increased formation of radiolabeled impurities over time. This included the hydrolyzed product 3-fluoro-5-(pyridin-2-ylethynyl)benzamide, which contributed to the low yields previously obtained for both the chloro- and nitro-precursors [13]. Our development of spirocyclic iodonium ylide (SCIDY) precursors for radiofluorination of non-activated aromatic rings [14] was successfully applied to the synthesis of (**5**) as a novel precursor for [^18^F]FPEB [13,15]. The ylide precursor showed a ten-fold increase in the radiochemical yield and a five-fold increase in molar activity (A_m_) of [^18^F]FPEB, compared with our traditional S_N_Ar reaction with the nitro-precursor [10], and was validated and translated for human use [12,13].

The aim of this study was to compare the chloro- and nitro-precursors for [^18^F]FPEB, with our first-generation SCIDY precursor (cyclopentyl auxiliary (5)) and new second-generation SCIDY precursor (adamantyl auxiliary (7)) and a newly reported sulfonium salt precursor (6) for the routine production of [^18^F]FPEB [16,17]. Only metal-free precursors were considered for this work to avoid the need for additional quality control testing of residual metals in routine radiopharmaceutical production. In light of our pilot PET imaging studies that showed increased [^18^F]FPEB binding in a patient with early mild cognitive impairment [10], we also explored the use of [^18^F]FPEB to detect early changes in mGluR5 expression in a transgenic murine model of AD.

**Table 1 molecules-25-00982-t001:** Reported Radiosyntheses of [^18^F]FPEB.

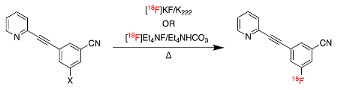					
**Compound**	**Precursor (X)**	**Reported [^18^F]FPEB Radiochemical Yields (RCYs)**	**Radiolabeling Method**	**Validated for Human Use**	**Reference**
**(1)**	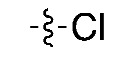	5%	Manual	✓	[9]
**(2)**	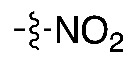	4–10%	Automated	✓	[8,10,18,19]
**(3)**	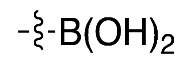	5% *	Automated		[11]
**(4)**	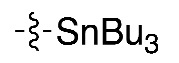	11% *	Manual		[12]
**(5)**	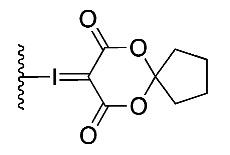	29%	Automated	✓	[13,15]
**(6)**	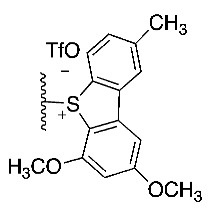	55%	Manual		[16]

* not isolated.

## 2. Results and Discussion

### 2.1. Radiosyntheses of [^18^F]FPEB with Five Different Precursors

The chloro-precursor (**1**) was synthesized following a literature procedure, with the modification that conventional heating was used to drive the radiofluorination instead of the previously reported microwave conditions [9]. In our hands, a maximum 3% RCC was observed by manual synthesis at 200 ℃ from a 10 min reaction. In light of the low radiofluorination conversion, this precursor was not considered for automated radiosynthesis (Table 2, Entry I). While manual radiosynthesis of [^18^F]FPEB via the commercially available nitro-precursor (**2**) showed an RCC based on radio-HPLC analysis of 33% (*n* = 3) at 150 °C for a reaction time of 5 min, upon automation and isolation, the highest RCY was only 4% (Table 2, Entry II). This low RCY from the nitro-precursor is consistent with other reported production yields [8,10,18,19]. Losses experienced when translated to an automated synthesis unit may be attributed to transfer to tubing and reaction vessels. The low yielding reaction, together with the presence of radiochemical and UV active impurities, observed by HPLC, led us to abandon the optimization of [^18^F]FPEB radiosynthesis using the nitro-precursor and explore alternative precursors.

We recently showed that using the SCIDY chemistry with a cyclopentyl auxiliary (SPI5) (**5**) as the precursor for [^18^F]FPEB production led to a five-fold increase in RCY and a three-fold increase in molar activity (A_m_) compared to the nitro-precursor. The SCIDY precursor enabled the displacement reaction with [^18^F]fluoride to be conducted at milder conditions (lower temperatures and shorter times), thereby minimizing the formation of radiochemical impurities. [^18^F]FPEB synthesis via the SCIDY SPI5 auxiliary precursor was conducted on a GE TRACERlab™ FX2 N automated synthesis module. Varying temperatures and reaction times resulted in RCYs of [^18^F]FPEB ranging from 19% to 23% after HPLC purification and formulation (Table 2, Entry III). [^18^F]FPEB production was repeated via the SPI5 precursor using the optimal conditions (100 °C, Et_4_NHCO_3_ as the base and a 5 min reaction time) which resulted in a RCY of 25% ± 2%, with a molar activity (Am) of = 37 ± 13 GBq/µmol (*n* = 3), consistent with our previous reports [13,15].

These automated reaction conditions were then applied to a novel SCIDY precursor for [^18^F]FPEB radiosynthesis, bearing an adamantyl-based auxiliary (SPIAd). Experimental and mechanistic studies have shown that radiofluorination with the bulkier SPIAd auxiliary in SCIDY precursors can improve the RCY compared to the SPI5 precursors in conventional nucleophilic aromatic substitution with [^18^F]fluoride [17,20]. The SPIAd precursor (compound **7**) also resulted in a 24% RCY for [^18^F]FPEB, and is similar to that of the SPI5 precursor (Table 2, Entry IV). Our initial semi-preparative HPLC purification conditions did not adequately separate the adamantyl precursor from [^18^F]FPEB, and given the equivalent RCY between the two different SCIDY auxiliaries, optimization of the SPIAd precursor reaction and/or HPLC conditions were not pursued (Appendix A).

A recently reported precursor (**6**) based on a sulfonium salt was also applied to the manual synthesis of [^18^F]FPEB and gave a reported 55% RCY [16]. Inspired by this report, we adapted the radiochemistry precursor and methodology for automated radiofluorination with the GE TRACERlab™ FX2 N synthesis module. The reaction was conducted at varying temperatures (80 or 100 °C) for 5 min and with varying bases (Et_4_NHCO_3_ or KHCO_3_/K_222_) to establish optimal temperature and base conditions, as summarized in Table 2, with the literature synthesis providing the highest yield [16]. Using [K_222_][^18^F] and the milder base (KHCO_3_) at 80 °C for 5 min (Table 2, Entry V), the sulfonium salt precursor produced [^18^F]FPEB with a RCY of 36% ± 6% and Am = 77 ± 35 GBq/µmol (*n* = 3).

When considering which chemical route should be used to produce [^18^F]FPEB for clinical research, many factors will impact this choice, including precursor availability, radiochemical yield, molar activity, ease of automation and purification, etc. Although the nitro-precursor is currently the only commercially available compound for [^18^F]FPEB production, the resulting yields are much lower than the SCIDY and sulfonium salt precursors. The SPI5 auxiliary and the sulfonium salt precursors appear to the be the best suited for routine radiopharmaceutical production of [^18^F]FPEB.

### 2.2. Small Animal PET/CT Imaging

mGluR5 has emerged as an imaging target in AD pathogenesis. It has been demonstrated that soluble oligomeric amyloid-β (Aβo) induces an accumulation and over-stabilization of mGluR5, and that Aβo up-regulates mGluR5, leading to an abnormal increase in the release of intracellular Ca^2+^ [6,21,22,23]. Preliminary PET imaging data with [^18^F]FPEB showed increased brain uptake in the transgenic model of AD versus the age-matched controls (10 month data shown in Figure 1). Dynamic PET/CT imaging was carried out to investigate the difference in [^18^F]FPEB binding between an established preclinical model of AD using 10 month old transgenic APP/PS1 mice and their age-matched wild-type (WT) controls. Axial, coronal, and sagittal images of the murine brains acquired at 20 min post-injection of [^18^F]FPEB are presented in Figure 1A. A marked increased uptake of radiotracer binding was observed in the brains of the transgenic mice compared with WT controls. The time–activity curves revealed a similar initial peak uptake and brain penetration for both groups of mice (Figure 1B). One minute after the time of injection (TOI), maximum uptake was observed in both genotypes, (SUV > 6; whole brain VOI). By 10 min post-injection, a modest higher [^18^F]FPEB retention was observed in the brain of transgenic mice (SUV = 4.8) versus WT controls (SUV = 4.0), and as the tracer cleared from normal tissues, a significant difference (*p* < 0.05) was apparent at <5 min post-injection. A comparison of the area under the curve (AUC) analysis (Figure 1C) over time for the respected genotypes underscores the trends observed in the TACs.

In a pilot study, we found that patients with early mild cognitive impairment had an increased brain uptake of [^18^F]FPEB, and that radiotracer uptake in the brain was reflective of increased mGluR5 density [10]. This observation supports the hypothesis that mGluR5 may be implicated in the early stages of AD pathogenesis [24]. Consistent with this hypothesis and clinical research indication, PET/CT imaging studies of [^18^F]FPEB uptake in a transgenic mouse model of AD also showed an increased radiotracer uptake and retention in the brain of the APP/PS1 mice, compared with wild-type controls. This preliminary work provides support that mGluR5 levels measured by [^18^F]FPEB are potentially useful as an early biomarker of AD. Further [^18^F]FPEB imaging studies and biological evaluations are underway including regional analysis of imaging data as well as ex vivo biodistribution and autoradiography studies to evaluate this functional link between mGluR5 and AD.

## 3. Materials and Methods

### 3.1. Materials and General Methods

Unless otherwise stated, all reagents were obtained from commercially available sources and used without further purification. The identification of all radiochemical products was determined by HPLC co-elution with an authentic non-radioactive standard. All RCC and RCY values are reported as decay corrected, relative to starting [^18^F]fluoride (ca. 100 mCi). [^18^F]Fluoride was produced using a Scanditronix MC17 cyclotron from enriched [^18^O]H_2_O through the ^18^O(p,n)^18^F reaction. Reverse-phase high performance liquid chromatography (HPLC) was used to isolate and purify [^18^F]FPEB. Dynamic PET/CT imaging experiments were conducted on a dedicated small-animal PET/CT scanner (eXplore *Vista-CT*, *Sedecal*, Algete, Spain) equipped with VISTA-CT version 4.11 software.

#### 3.1.1. General Chemistry and Radiochemistry Methods

Preparation of the chloro-precursor was performed as previously reported [9]. The nitro-precursor (Lot No.: 20190401) and FPEB standard (Lot No.: 20130101) were purchased from ABX. Synthesis of the sulfonium salt precursor and radiolabeling were performed as previously reported [16]. For manual labeling, azeotropically dried potassium cryptand [^18^F]fluoride was dissolved in DMSO (3.0 mL), while 400 μL aliquots (typically 3–5 mCi) were used per reaction to dissolve the precursor in a 1 dram vial. The reaction was heated at 150 °C for 5 min, then quenched with water and cooled for 3 min. Product identity and radiochemical conversion were determined as the ratio of free [^18^F]fluoride to [^18^F]FPEB as integrated by radio-HPLC. The manual radiolabeling of the chloro- (**1**) and nitro- (**2**) precursors was performed as previously reported, with slight modification to the concentration and reaction time. One milligram of **2** was dissolved in 0.4 mL [^18^F]fluoride in DMSO for 5 min instead of 1.5 mL for 15 min [8,18], and the chloro-reaction was conducted at high temperatures instead of under microwave conditions [9]. For automated radiosyntheses, a GE TRACERlab™ FX2 N module with **2** and the SPI5 auxiliary SCIDY precursor (**5**) were performed as previously reported [13,15]. Flash chromatography was performed on a Biotage Isolera One automated flash purification system. Biotage SNAP KP-Sil 50 g cartridges (45–60 micron) were used with a flow rate of 50 mL/min for gradient solvent systems. Fractions were monitored and collected by UV absorbance using the internal UV detector set at 254 and 280 nm.

#### 3.1.2. SCIDY-SPIAd Auxilary Synthesis and Characterization

The titled compound was prepared using a modified literature procedure [15]. Trifluoroacetic acid (0.9 mL) was added to a solution of IPEB (120 mg, 0.36 mmol) in chloroform (0.12 mL). Oxone (179 mg, 0.58 mmol) was added and the reaction mixture was stirred for 3 h, until full conversion of starting materials was determined by TLC (SiO_2_ coated on polyethylene, 250 μm, with 100% EtOAc). Volatile contents were then removed by rotary evaporation. The round bottom flask was covered in foil and further dried under high vacuum for 5 h. The dried residue was suspended in ethanol (1.5 mL) and (1*r*,3*r*,5*r*,7*r*)-spiro[adamantane-2,2′-[1,3]dioxane]-4′,6′-dione (67 mg, 0.54 mmol). SPIAd was added followed by 10% Na_2_CO_3_(aq) (*w*/*v*, 1.5 mL, 0.33 M solution) in ~0.2 mL aliquots until the pH of the reaction mixture was equal to pH 10. The reaction mixture was stirred for 5 h until full conversion to the iodonium ylide was determined by TLC (SiO_2_ coated on polyethylene, 250 μm, with 10% EtOH in EtOAc, (1:9 mL *v*/*v*)). The reaction mixture was then diluted with water and extracted with chloroform. The chloroform extracts were combined and washed with water (4 × 10 mL) and brine (1 × 10 mL). The organic layer was dried with anhydrous MgSO_4_, filtered, and concentrated. The final compound was purified by flash chromatography using a gradient 60% EtOAc in hexanes to 100% EtOAc to 5% methanol in EtOAc. Compound **7** (56 mg, 0.11 mmol) was isolated as an off-white powder with a 41% yield.

^1^H NMR (500 MHz, dmso-d6) δ (ppm): 8.63 (dt, *J* = 4.7, 1.3 Hz, 1H), 8.31 (t, *J* = 1.5 Hz, 1H), 8.25 (t, *J* = 1.6 Hz, 1H), 8.17 (t, *J* = 1.6 Hz, 1H), 7.89 (td, *J* = 7.7, 1.8 Hz, 1H), 7.73–7.67 (m, 1H), 7.47 (ddd, *J* = 7.7, 4.8, 1.2 Hz, 1H), 2.35 (s, 2H), 1.93 (d, *J* = 12.3 Hz, 5H), 1.80–1.75 (m, 2H), 1.68–1.62 (m, 7H). ^13^C NMR (126 MHz, dmso-d6) δ (ppm):163.10, 150.88, 141.60, 139.17, 137.54, 137.49, 136.20, 128.34, 125.05, 124.86, 117.02, 116.90, 114.37, 105.94, 92.58, 84.86, 59.00, 36.92, 35.31, 33.64, 26.38. HRMS (m/z): [M + Na]^+^ calcd. for C_22_H_15_IN_2_O_4_Na, 520.9974; found 520.9967.

#### 3.1.3. General GE TRACERlab™ FX2 N Automated Synthesis Method

Cyclotron produced [^18^F]fluoride in enriched [^18^O]H_2_O was delivered to a GE TRACERlab™ FX2 N automated synthesis module. The [^18^F]fluoride was trapped on a HCO_3_^−^ anion exchange cartridge (Chromafix^®^ [^18^F] SPE) without additional conditioning and eluted with one of the following base mixtures: Et_4_NHCO_3_ in CH_3_CN/H_2_O, 4,7,13,16,21,24-hexaoxa-1,-10,diazabicyclo [8.8.8]hexacosane (Kryptofix 222) and KHCO_3_ in CH_3_CN/H_2_O or Kryptofix 222 and K_2_CO_3_ in MeOH/H_2_O. The [^18^F]fluoride was released from the cartridge and dried under nitrogen gas at 90 °C, followed by azeotropic drying with acetonitrile under nitrogen gas at 110 °C, producing either dry [^18^F]Et_4_NF or [^18^F]KF/K_222_ for radiofluorination. Automated radiosyntheses were carried out with precursors **2**, **5**, **6**, and **7**. Each precursor was dissolved in either DMSO, DMF, or CH_3_CN and eluted into the reaction vial with the dry [^18^F]fluoride, where they were heated for the specified reaction time at varying temperatures. The reactions were then cooled to room temperature and quenched with HPLC buffer. The mixture was diluted with water and the crude reaction mixture was injected onto a semi-preparative HPLC column. The eluent was monitored by UV, at a wavelength of 254 nm, and radiochemical detectors in series. The desired radiochemical product was collected and formulated by dilution with sterile water and loaded onto a C18 Sep-Pak cartridge (pre-activated with 10 mL EtOH, followed by 10 mL H_2_O). The cartridge was washed with water to remove impurities and then eluted with dehydrated EtOH, and finally, diluted with 0.9% sodium chloride. The final radiochemical yield of this formulated product was determined (50–60 min; see ESI). Molar activity measurements were carried out as described in [12].

#### 3.1.4. Preliminary Small Animal PET/CT Imaging Studies

All animal experiments were conducted in compliance with Institutional Animal Care and Use Committee (IACUC) guidelines and the Guide for the Care and Use of Laboratory Animals. Female wild-type B6C3F1/J mice and female transgenic B6C3-Tg(APPswe,PSEN1dE9)85Dbo/J (APP/PS1; Stock No.: 034829-JAX or MMRRC No. 034) mice were obtained from Jackson Laboratory (Bar Harbor, ME). Mice were provided with food and water ad libitum. Mice were subjected to PET/CT imaging studies after being aged to 10 months. Mice were administered formulations of [^18^F]FPEB (~3.1–14.0 MBq [~84–378 µCi], specific activity of 5 Ci/μmol, in 200 µL sterile PBS, pH7.4, ≤5% *v*/*v* EtOH) via intravenous (i.v.) tail-vein injection using a catheter. Approximately 5 min prior to recording PET images, the mice were anesthetized by inhalation of 3–4% isoflurane (Baxter Healthcare, Deerfield, IL)/oxygen gas mixture, and a catheter was inserted into the tail-vein, with the mice then transferred to the scanner bed and placed in the prone position. Anesthesia was maintained with 1–2% isoflurane/oxygen gas mixture (flow rate ~5 L/min). Co-registered dynamic PET/CT images were recorded for a total of 20 min post-injection radiotracer injection (*n* = 3 per group). List-mode data were acquired for 20 min per scan using a γ-ray energy window of 250–700 keV. To ensure that the activity bolus was measured, PET/CT data acquisition was initiated 20 s prior to injecting the radioactivity. Data were processed by 3-dimensional Fourier re-binning (3D-FORE), and images were reconstructed using the 2-dimensional ordered-subset expectation maximum (2D-OSEM) algorithm. Image data were normalized to correct for the non-uniformity of response of the PET, dead-time count losses, positron branching ratio, and physical decay to the time of injection, but no attenuation, scatter, or partial-volume averaging correction were applied. An empirically determined system calibration factor (in units of Bq/cps) combined with the decay corrected administered activity and the animals’ weights were used to parameterize image activity in terms of the standardized uptake value (SUV). Manually drawn 2-dimensional regions-of-interest (ROIs) or 3-dimensional volumes-of-interest (VOIs) were used to determine the maximum and mean SUV radiotracer uptake in various tissues. Time–activity curves (TACs) were generated from the ROI analysis on dynamic PET/CT data using 20 s frames. CT images were recorded using an X-ray current of 300 μA, 360 projections, and an image size of 63.8 mm × 63.8 mm × 46.0 mm. Data were acquired using the Vista CT 4.11 Build 701 software, and reconstructed images were analyzed by using ASIPro VM^TM^ software (Concorde Microsystems, Siemens Preclinical Solutions, LLC, Knoxville, TN, USA) and VivoQuant^®^ 1.23 (InviCRO, LLC, Boston, MA, USA).

### 3.2. Data Analysis and Statistics

Data and statistical analyses were performed using GraphPad Prism 5.01 (GraphPad Software, Inc., La Jolla, CA, USA) and Microsoft Excel spreadsheets. Differences at the 95% confidence level (*p* < 0.05) were considered to be statistically significant.

## Figures and Tables

**Figure 1 molecules-25-00982-f001:**
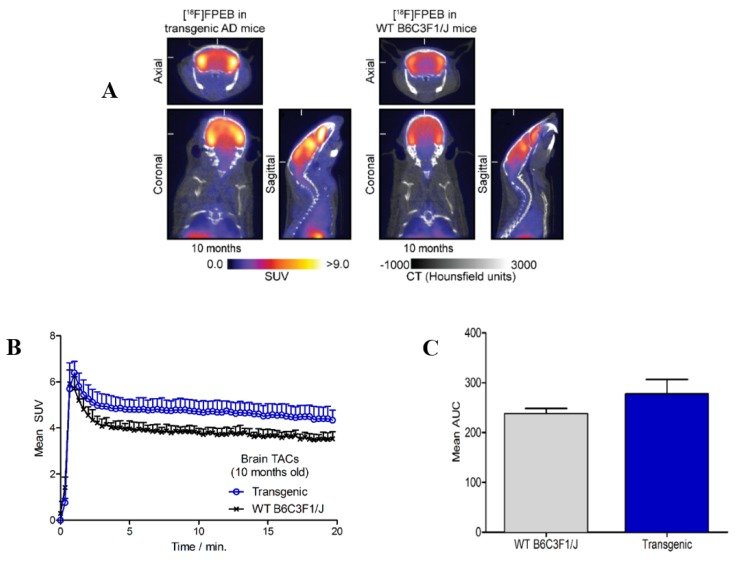
(**A**) PET/CT images of [^18^F]FPEB 20 min post-injection in 10 month old APP/PS1 (transgenic) mice and aged-matched wild-type (WT) B6C3F1/J (control) mice, *n* = 3/group. (**B**) Time–activity curve for whole-brain. (**C**) Mean area under the curve (AUC).

**Table 2 molecules-25-00982-t002:** Summary of automated radiosyntheses of [^18^F]FPEB (this work).

Entry	Precursor (X=)	Base	Solvent	Temperature (°C)	Reaction Time (min)	RCY (%)	Am (GBq/µmol)
I	(**1**)	K_2_CO_3_/K_222_	DMSO	200	10	N/A	N/A
II	(**2**)	K_2_CO_3_/K_222_	DMSO	150	5	4	N/A
III	(**5**)	Et_4_NHCO_3_	DMF	80	5	23	22
		Et_4_NHCO_3_	DMF	80	10	19	54
		Et_4_NHCO_3_	DMF	100	5	25 ± 2 *	37 ± 13 *
IV	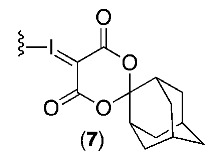	Et_4_NHCO_3_	DMF	100	5	24	21
V	(**6**)	Et_4_NHCO_3_	CH_3_CN	80	5	15	70
		Et_4_NHCO_3_	CH_3_CN	100	5	26	168
		KHCO_3_/K_222_	CH_3_CN	80	5	36 ± 6 *	77 ± 35 *

* *n* = 3.

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
