# Peer review of "Revisiting the Radiosynthesis of [18F]FPEB and Preliminary PET Imaging in a Mouse Model of Alzheimer’s Disease"

_molecules, 2020, doi:10.3390/molecules25040982_

Round 1

Reviewer 1 Report

The manuscript molecules-720247 describes the investigation on different protocols for labeling of [18F]FPEB and its preliminary PET imaging in an AD mouse model. Five labeling precursors were compared with regard to RCY and molar activity of product. In addition to already known compounds (1, 2, 5 and 6, numbers are missing in table of scheme 1) a new iodonium ylide (compound 7) was synthesized and applied.

The results obtained were, although not really surprising, interesting and probably of future value, in view of the confirmed excellent data gained with compound 6. However, what is about using a Bpin precursor as described in Preshlock et al. (Chem. Commun. 2016, 52, 8361)?

The in vivo evaluation of the tracer in App/Ps1 mice vs. WT as control is well presented. The reference [13] given in regard to [18F]FPEB synthesis seems to be unsuitable since this tracer is not mentioned in this publication.

The paper can be accepted after minor revision.

Specifically, please check the following in lines:

Line 67 square brackets: exchange “11-13” by “11, 12”, correct?

Line 70 square brackets: remove “13” or exchange “13” by “12”

Line 76 Table 1, accordingly in line for compound (5), exchange “[11-13]” by “[11, 12]”;

In column “Precursor (X)” the attachment point given for compounds (5) and (6) should be also applied for comp. (1) – (4), otherwise it is just a leaving group.

Line 128 Scheme 1 (Table), in column “Precursor (X=)” accordingly use attachment points for precursors entry I and II. Numbers of precursor comp. are missing for entries I, II, III and V.

Line 201 Remove “of”

Line 204 please check MgSO4 (use subscript)

Line 205 Please check, “hexanes” is not usually capitalized

Line 206 check “EtOAC” vs “EtOAc”

Line 219 please check “Krypotfix”

Line 249 “2-dimensional”

Reviewer 2 Report

A well written and clear article. Although much of the chemistry has been previously reported, including the use of spirocyclic iodonium ylide and sulfonium salt precursors for the synthesis of [18F]FPEB, the articles very nicely outlines optimization conditions and provides validation of previously reported yields in the literature. This is useful for the community. Addition of total synthesis time and/or non-decay corrected yields as well details on molar activity measurement method, including HPLC traces, would be helpful.

The in vivo results although preliminary, seem promising. Given the slight differences between the WT and APP/PS1 mice, and the inherent variability in manually drawn regions of interest, further validation studies are recommended before definitive conclusions can be drawn.  Further supportive data may include ex-vivo counting of brain uptake, use of specific and reference regions in addition to whole brain measurement, and ex vivo staining for mGluR5 and auto radiography correlation.
